# Discovery of Small-Molecule Activators for Glucose-6-Phosphate Dehydrogenase (G6PD) Using Machine Learning Approaches

**DOI:** 10.3390/ijms21041523

**Published:** 2020-02-23

**Authors:** Madhu Sudhana Saddala, Anton Lennikov, Hu Huang

**Affiliations:** Mason Eye Institute, University of Missouri School of Medicine, Columbia, MO 65201, USA; saddalam@missouri.edu (M.S.S.); lennikova@missouri.edu (A.L.)

**Keywords:** G6PD, pharmacophore modeling, machine learning, docking, ADMET

## Abstract

Glucose-6-Phosphate Dehydrogenase (G6PD) is a ubiquitous cytoplasmic enzyme converting glucose-6-phosphate into 6-phosphogluconate in the pentose phosphate pathway (PPP). The G6PD deficiency renders the inability to regenerate glutathione due to lack of Nicotine Adenosine Dinucleotide Phosphate (NADPH) and produces stress conditions that can cause oxidative injury to photoreceptors, retinal cells, and blood barrier function. In this study, we constructed pharmacophore-based models based on the complex of G6PD with compound AG1 (G6PD activator) followed by virtual screening. Fifty-three hit molecules were mapped with core pharmacophore features. We performed molecular descriptor calculation, clustering, and principal component analysis (PCA) to pharmacophore hit molecules and further applied statistical machine learning methods. Optimal performance of pharmacophore modeling and machine learning approaches classified the 53 hits as drug-like (18) and nondrug-like (35) compounds. The drug-like compounds further evaluated our established cheminformatics pipeline (molecular docking and *in silico* ADMET (absorption, distribution, metabolism, excretion and toxicity) analysis). Finally, five lead molecules with different scaffolds were selected by binding energies and *in silico* ADMET properties. This study proposes that the combination of machine learning methods with traditional structure-based virtual screening can effectively strengthen the ability to find potential G6PD activators used for G6PD deficiency diseases. Moreover, these compounds can be considered as safe agents for further validation studies at the cell level, animal model, and even clinic setting.

## 1. Introduction

Glucose-6-Phosphate Dehydrogenase (G6PD) is a ubiquitous cytoplasmic enzyme converting glucose-6-phosphate into 6-phosphogluconate in the pentose phosphate pathway (PPP). The PPP products are crucial for the biosynthesis of nucleotides and fatty acids. G6PD is the first and rate-limiting enzyme of this metabolic pathway that supplies reducing energy to cells by maintaining the level of the reduced form of the extramitochondrial Nicotine Adenosine Dinucleotide Phosphate (NADPH) coenzyme [1]. NADPH enables cells to counterbalance oxidative stress that can be prompted by several oxidant agents and to preserve the reduced form of glutathione. The PPP is the dominant source of NADPH; therefore, defense against oxidative damage is dependent on G6PD [2]. NADPH serves as an electron donor for various enzymatic reactions essential in biosynthetic pathways, and its production is crucial to the protection of cells from oxidative stress. G6PD is also necessary to regenerate the reduced form of glutathione that is produced with one molecule of NADPH. The reduced form of glutathione is essential for the reduction of hydrogen peroxide to balance oxidative stress [3]. The monomer of G6PD consists of 515 amino acids, with a molecular weight of about 59 kDa. A model of the three-dimensional structure of G6PD was reported in 1996 [4], and, subsequently, the crystal structure of human G6PD was elucidated [5]. The enzyme is active as a tetramer or dimer in a pH-dependent equilibrium. Every monomer consists of two domains: the N-terminal domain (amino acids 27–200), with a β–α–β dinucleotide binding site (amino acids 38–44); and a second larger β+α domain, consisting of an antiparallel nine-stranded sheet. The dimer interface lies in a barrel arrangement in this second part of the G6PD molecule. The two domains are linked by an α helix, containing the totally conserved eight-residue peptide that acts as the substrate-binding site (amino acids 198–206) [6]. G6PD is the only enzyme known to have evolved a second NADP+ binding site, close to the dimer interface; this second structural site is essential for maintaining the activity, stability, and oligomeric state of the enzyme [7]. 

G6PD deficiency is the most common human enzyme defect, present in more than 400 million people worldwide [8]. A lack of G6PD leads to a shortage of NADPH and the inability to regenerate glutathione, leaving red blood cells (RBCs) susceptible to oxidative injury. This is the predominant mechanism that leads to erythrocyte lysis and intravascular hemolysis. Predominantly in the retina, G6PD deficiency may accelerate the dystrophic conditions (e.g., retinitis pigmentosa) to the photoreceptor cells of the retina [9]. In G6PD deficient patients, the inability to eliminate oxidants as a result of lower levels of reduced glutathione contributes to the oxidative injury of the retinal photoreceptor cells [10,11]. Interestingly, deficiency of G6PD has been associated as a risk factor for proliferative diabetic retinopathy (pDR) in patients with type 1 diabetes [12]. Recently, we showed that recombinant PIGF (placental growth factor) and PGF/VEGF heterodimers reduced expression of G6PD to mediate diabetes-induced BRB (blood-retinal barrier) breakdown via interaction with NFκB (nuclear factor kappa-light-chain-enhancer of activated B cells) p65 (manuscript under review). We also reported that PIGF blockade by antibody (ab) enhanced expression of G6PD and promoted the antioxidant defense mechanism [13]. Hence, discovering novel potential scaffolds is still in great demand, and we will be committed to discovering novel potent G6PD activators.

The aim of the current study is to design a systematic approach for the discovery of novel activators by searching publicly available databases. The structure-based pharmacophore model was generated with the help of the LigandScout software. This pharmacophore model was used as a 3D query to search novel compounds using the PubChem database. Following hits selection, the hits are subjected to machine learning, molecular docking, protein–ligand interaction, and *in silico* ADMET studies. These G6PD small-molecule activators will be promising cures for G6PD deficiency diseases through boosting the enzymatic activity of the remaining G6PD protein. 

## 2. Results

### 2.1. Target Protein Preparation

The crystal structure of the G6PD complex with NADP^+^ obtained through the X-ray diffraction method, 1.90 Å resolution, 0.213 R-value free, and 0.172 R-value work were downloaded from the protein data bank (PDB). All of the heteroatoms, crystal water molecules, and default ligand molecules (NADP and Glycerol) were removed from the target protein. We performed energy minimization to G6PD protein by MOE algorithms. During the minimization process, only torsion angles in the side chains of amino acids were modified, and other properties including bond lengths and backbone atom positions were held fixed. For simulating in vivo solvent conditions, we performed molecular docking operation in the presence of water and metal ions. Water box and neutralizing ions were added to G6PD protein by web-based CHARMM (Chemistry at Harvard Macromolecular Mechanics) algorithm (http://www.charmm-gui.org/). The refined G6PD has various binding sites like two NADP binding sites, one substrate-binding site, and a dimer interface binding site, respectively. 

The CASTp results showed that the pocket id is 3, area (SA) is 91.819, and volume (SA) is 66.926, respectively. The G6PD structure showed that the active human G6PD enzyme exists in a dimer or tetramer equilibrium. The dimeric G6PD enzyme has two subunits symmetrically located across a complex interface of β-sheets, and each subunit binds to a nicotinamide adenine dinucleotide phosphate (NADP^+^) molecule that confers structural stability. This structural NADP+ molecule is positioned close to the interface where the two subunits of each dimer are intertwined. The active sites of G6PD protein are presented in Figure 1. These active site residues play a beneficial role in the pathological consequences of the dysregulated G6PD signaling in the pentose phosphate pathway. 

### 2.2. Pharmacophore Modeling

The pharmacophore model comprises several features organized in a specific 3D pattern. Each feature is typically represented as a sphere. Such pharmacophore features are typically used as queries to screen small molecule libraries of compounds. Here, we took the G6PD complex with AG1 that was assigned to the Ligandscout v4.3, Austria, for modeling pharmacophore models. The LS generated G6PD pharmacophore has four pharmacophoric features with AG1 ligands, such as two hydrogen bond donors (HBD) (green), one positive ionization sphere (light blue), and one aromatic ring (dark blue), respectively (Figure 2A). The HBD sphere1 (green) has a 1.38 radius, along with x = 28.61, y = 35.62, z = 62.55, θ = 137.979, and φ = −108.402. The HBD sphere2 (green) has a 1.36 radius, along with x = 28.94, y = 34.57, z = 61.79, θ = 134.250, and φ = −105.978. The positive ionization of ligand (NH2 function group) interacted with GLU-419 amino acid, the first HBD interacted with ASP-421 amino acid, and the second HBD interacted with HIS-513 amino acid, respectively. The four pharmacophoric features play a vital role in screening the small molecules from PubChem libraries (Figure 2B).

### 2.3. Pharmacophore Based Virtual Screening

Pharmacophore screening is a drug discovery approach that employs a spatial arrangement of chemical groups of a ligand within the receptor, usually the target protein. This approach is commonly performed computationally and is arguably one of the most established and effective methods of virtual screening for rational drug design and discovery [14]. The pharmacophore-based screened results showed that the screening generated 938 unique hits selected from the PubChem database. Hits were filtered by the Pan-Assay Interference compounds (PAINS)–Remover engine [15]. The 442 hits were further filtered by setting 1.2 Å as the maximum RMSD to restrict the hits to those that have the best overall geometric match to the pharmacophore hypothesis. Finally, 53 best pharmacophore hits were considered for further studies.

### 2.4. Molecular Descriptors’ Calculation and Clustering

Predictions of small molecule physicochemical properties are important for assessing their ‘drug-likeness’ and ‘lead likeness’ in silico [16]. They are also useful for enriching compound collections with desirable properties. Physicochemical property data are essential for predicting bioactive and other properties of small molecules using modern machine learning approaches [17]. ChemmineR calculated the 38 physicochemical property values, including Lipinski descriptors for custom compound sets by JOELib package. The resulting properties were further used to cluster compounds by similar property profiles and visualized as a heat map (Figure 3A) hierarchical clustering trees and Principle Component Analysis (PCA). The hierarchical clustering showed seven cluster trees. Cluster 1 has four compounds, Cluster 2 and Cluster 3 have ten compounds each, Cluster 4 has fourteen compounds, Cluster 5 has four compounds, Cluster 6 has five compounds, and Cluster 7 has seven compounds, respectively (Figure 3B). The PCA showed various groups of compounds based on the Tanimoto coefficient (distance) between the first component (PCA1) against the second component (PCA2). The logarithm of the calculated partition coefficient (logP) against the polar surface area (PSA) showed that the compounds have a maximum of 5.8 logP and 66 PSA. The molecular weight (MW) against the PSA showed that the compounds have a maximum of 66 PSA and 400 MW. The molecular weight (MW) against the logarithm of the calculated Partition coefficient (logP) showed that the compounds have a maximum of 5.8 logP and 400 MW (Figure 4). These results revealed that the 53 compounds have various molecular properties and different functional groups.

### 2.5. Statistical Machine Learning Methods

The training set (311 drug-like and 320 nondrug-like compounds) and testing set (53 hits) were used as input datasets to five statistical machine learning methods. According to Korkmaz et al. [18] six molecular descriptors as the best features were calculated, including logP, polar surface area (PSA), donor count (DC), aliphatic ring count (AlRC), aromatic ring count (ArRC), and Balaban index (BI) to test set (see Appendix A). The atom-additive XLOGP method is used to calculate the logP, and other descriptors are calculated by using the Marvin Beans tool and ChemmineR package. We used test data (53 hits) sets with the best six features to the above-mentioned machine-learning methods. Then, we applied the same training procedure. The results were as follows: for FDA, product degree, and number of terms are acquired as 1 and 7, respectively; for C5.0, number of boosting iterations is selected as 10, and a tree-based model is used, whereas predictor winnowing is not used; for J48, confidence threshold is set as 0.25, for lsSVMrbf, optimal sigma parameter is obtained as 0.27, for SVMrbf, sigma, and cost parameters are determined as 0.30 and 1, respectively, for SVMlin, cost parameter selected as 1, for RF, number of randomly selected predictors set as 2 and 500 trees are used, for bagSVM, number of bootstraps are set as 100 and radial basis function used as kernel, for NN, number of hidden units and weight decay are optimized as 19 and 0.1, respectively, and, for KNN, number of neighbors are selected as 11. The suggested statistical machine learning methods classified results immediately as drug-like or nondrug-like for each compound. The machine learning that resulted showed that among 53 hit compounds, 18 compounds were drug-like, and 35 compounds nondrug-like (Figure 5). The drug-like 18 compounds were used for further molecular docking studies.

### 2.6. Molecular Docking

Molecular docking is a computational technique that illustrates the conformations of compounds in protein binding sites. Docking was performed using the AutoDock in the PyRx Virtual Screening tool. The 53 hits were proposed to bind with G6PD within the active pocket (dimer interface) (Figure 6). Based on the binding conformation, AutoDock generated binding energies for all molecules. We have selected the top five compounds based on the best binding energies and performed G6PD-small molecule interactions. We illustrated the SMILES notation, bind energies, and interacted amino acids of five compounds (Table 1). The protein–small molecule interaction results showed that the CID6917760 compound bound (ne–8.9 kcal/mole) active site of G6PD (dimer interface domain) and interacted with ILE-220, PHE-221, ASN-229, ASN-388, ILE-224, PHE-373, TYR-401, VAL-400, THR-402, ASP-421, LEU-420, and LEU-422 active site residues. CID9820229 compound bound (ne–7.6 kcal/mol) active site of G6PD and interacted with LEU-214, PHE-221, LEU-420, THR-402, ILE-220, PHE-373, ASN-388, ILU-224, and LEU-422 active site residues. CID5221957 compound bound (ne–7.3 kcal/mol) active site of G6PD and interacted with ASP-421, LEU-422, TYR-401, LEU-420, VAL-400, THR-402, PHE-373, ILE-224, ASN-388, HIS-374, ASP-375, ASN-229, VAL-376, PHE-221, and ILE-220 active site residues. CID389556 compound bound (ne–7.2 kcal/mol) active site of G6PD and interacted with LEU-420, VAL-400, ASP-421, TYR-401, THR-402, LEU-422, PHE-373, ASN-388, ILE-224, PHE-221, and ILE-220 active site residues. CID10900930 compound bound (ne–7.0 kcal/mol) active site of G6PD and interacted with PHE-373, THR-402, LEU-422, TYR-401, ASP-521, LEU-420, VAL-400, PHE-221, ILE-220, and ASN-388 active site residues. AG1 (CID6615809) bound (ne–6.1 kcal/mol) active site of G6PD and interacted with and interacted with LEU-420, THR-423, ASN-426, ASP-421, ARG-427, and LEU-422 active site amino acids (Figure 7). The molecular docking results revealed that all of the top five compounds were bound exactly in the AG1 binding site position and interacted with the functional residues. The protein–ligand interaction results suggested that five compounds bound (dimer interface domain) stronger than the AG1 molecule and may increase the expression of G6PD signaling in the pentose phosphate pathway.

### 2.7. Pharmacokinetic and Toxicity Risks Assessment

The pharmacokinetic properties (ADMET) are of prime importance for a molecule eligible to be an active drug. The poor ADMET properties often lead to the failure of an otherwise potent drug. Thus, it is crucial to investigate the pharmacokinetic profile of a potential drug as well. Therefore, in order to evaluate the druggability of best five compounds using Lipinski’s rule of five, the physically important descriptors and pharmaceutically relevant ADMET properties were evaluated using the SwissADME tool and Osiris molecular property explorer. 

The molecular complexity of the five compounds could be measured by the number of rings and aromatic rings, the fraction of carbons that were sp3 hybridized (Csp3), or the number of stereocenter properties and ADMET properties, which were all computed by SwissADME tool (Table 2). All the compounds had a good bioavailability score; consensus predicted logP, molar refractivity (MR), molecular weight (MW), synthetic accessibility, topological polar surface area (TPSA), GI (gastrointestinal) absorption, BBB (blood–brain barrier) permeant, CYP1, CYP2, CYP3 inhibitor, and lead likeness, respectively (Table 3). We are also found that overall all the compounds followed the Lipinski’s rule of five (below 5 hydrogen bond donors and 10 hydrogen bond acceptors, 500 molecular weight (MW) and 5 logP (n-octanol and water partition coefficient)) except CID6917760 compound (1 violation, MLOGP>4.15), but it might be acceptable. The rest of the active compounds followed Lipinski’s rule and have reliable polarity for better permeation and absorption as revealed by H-bond donors and H-bond acceptors (Figure 8). The compounds were also followed the Ghose rule, Veber rule, and Egan rule (Table 2). The pharmacokinetic profile results revealed that all the compounds had followed the good drug molecule properties and were acceptable. 

We are also calculated the toxicity risk assessment properties like mutagenicity, tumorigenicity, irritation effect, and the risk of reproductive effect were predicted. The Osiris toxicity risk predictor locates fragments within a molecule, which indicates a potential toxicity risk. The toxicity risk assessment results showed that compounds for all five of the compounds have no risk of mutagenicity, tumorigenicity irritation, and reproductive toxicity shown in Table 3. The CID10900930 compound is partially mutagenic and tumorigenic. To assess the lead’s overall potential to qualify for a drug, we calculated the overall drug score, which combines drug-likeness, hydrophilicity (cLogP), aqueous solubility (LogS), MW, and toxicity risk parameters. The logP value was predicted to determine the hydrophilicity of all compounds. The results of toxicity risk assessment screening revealed an overall drug score of predicted active compounds. This result further reassures us to discover newer G6PD small-molecule activators for the G6PD deficiency’s diseases.

## 3. Discussion

G6PD deficiency is an X-linked, hereditary genetic defect caused by mutations in the G6PD gene, resulting in protein variants with different levels of enzyme activity. [19] The G6PD deficiency causes a variety of pathologies, including kidney injury, heart failure, psychiatric disorder, diabetes, cholelithiasis, and cataracts, which accelerates the dystrophic conditions to photoreceptor cells and damages the retinal blood barrier function. The most common clinical manifestations are neonatal jaundice and acute hemolytic anemia, which in most patients is triggered by an exogenous agent. In the retina, the G6PD deficiency can create stress conditions that can damage photoreceptor, retinal cells, and blood barrier function, leading to retinal diseases, such as photoreceptor degeneration and diabetic retinopathy. These wide spectrums of pathologies are corroborated by the fact that G6PD deficiency is a highly prevalent risk factor for multiple human diseases [20,21,22,23,24]. 

In the present study, we screened the novel hit molecules for G6PD protein dimerization from the PubChem library using pharmacophore-based virtual screening followed by statistical machine learning methods, molecular docking, and *in silico* ADMET studies. The identified novel small-molecule activators may provide treatment options for the patients with the G6PD enzymopathies, affecting more than 400 million people worldwide. It could be expected that many other pathologies associated with G6PD deficiency, as aforementioned, are trailed by the identified five novel small-molecule activators. In addition, they may also be valuable to G6PD-deficiency populations in developing countries, where hemolytic crisis-triggering factors (e.g., infection, food, and drugs) are still common. Hwang et al. [6] identified AG1, a small molecule that increases the activity of the wild-type, the Canton mutant, and several other common G6PD mutants and reduces oxidative stress in cells and zebrafish. Furthermore, AG1 decreases chloroquine or diamide-induced oxidative stress in human erythrocytes. Raub et al. [7] studies reported small-molecule activators of G6PD bridging the dimer interface at the structural nicotinamide adenine dinucleotide phosphate (NADP^+^) binding sites of two interacting G6PD monomers. Genetic mutation causes reduced efficiency of the dimerization of G6PD monomers affecting its biological activity; in many cases, the pathological effects of G6PD deficiency were not associated with the reduced expression, but reduced binding energy affecting formation of G6PD homodimers (active form), such as class I G6PD variants, associated with chronic nonspherocytic hemolytic anemia (CNSHA), the most severe phenotypic expression of G6PD deficiency [25]. The predicted compounds were expected to overcome reduced native binding of pathological forms of G6PD protein acting as a chemical substitute and forcing G6PD dimerization and restoring its biological role in the cells. In terms of the statistical machine learning methods, docking scores, and in silico ADMET studies, the results were satisfactory, indicating that the virtual screening strategy combined with machine learning as well as structure-based molecular docking may improve the efficiency and the accuracy of finding active target compounds in drug design and discovery. 

Our current effort focuses on improving the biochemical and pharmacological features of five novel small-molecule activators identified through further medicinal chemistry efforts and structural studies. Finally, identifying small-molecule enzyme activators is still considered a challenging task in the field of drug discovery and development. Further preclinical studies in G6PD deficient model organisms such as zebrafish [26] and mice [27] required fully elucidating therapeutic potential and effective dosage of the identified compounds.

## 4. Materials and Methods

### 4.1. Preparation of Target Protein

The crystal structure of G6PD in complex with structural NADP (PDB ID: 2BHL) was downloaded from the Protein Data Bank (PDB) (https://www.rcsb.org/structure/2BHL) [6] (Appendix A). In addition, water molecules and default ligands that are not involved in binding were removed according to the reported interactions for protein structure (Appendix A). The standard MOE v.2018, USA, (Molecular Operating Environment) software (https://www.chemcomp.com/) was utilized for rendering the missing charges, protonation states, and assigning of polar hydrogen to the G6PD protein. 

### 4.2. Active Site Prediction 

The CASTp (Computed Atlas of Surface Topography of proteins) server (http://sts.bioe.uic.edu/castp/index.html?2pk9) was used to predict the G6PD active sites [28,29,30] Appendix A. CASTp measures and identifies pockets and pocket mouth openings, in addition to the cavities. We uploaded the G6PD protein as input to predict the ligand binding active sites. The CASTp server predicted the key amino acids for binding interactions to the inhibitor/activator [31]. 

### 4.3. Generation of the Structure-Based Pharmacophore Model

The Structure-Based pharmacophore model was generated using the G6PD-AG1 (G6PD activator AG1) complex (Appendix A). The ligand–protein interaction residues’ chemical features were selected. The G6PD complex with AG1 was assigned to the Ligandscout v4.3, Austria, (LS) for pharmacophore modeling. LS could be an apparatus that permits the automated construction in addition to the apparition of 3D pharmacophore as of structural knowledge of macromolecules⁄inhibitor complexes [32]. We applied LigandScout software for generating the pharmacophore model, which contains all the chemical features information such as two hydrogen bond donors (HBD), one aromatic ring, and one positive ionization within the binding site sphere of the G6PD protein.

### 4.4. Pharmacophore Based Virtual Screening

We designed the structure-based pharmacophore model, searched against the commercially existing PubChem database using LigandScout software for matching pharmacophore features. The four pharmacophore features acted as query parameters along with hit reduction, hit screening, and subset selection, which we set as follows: Max Hits per Conformation was set to 1, Max Hits per Molecular was set to 1, Max Total Hits was set to default, Max RMSD (root mean square deviation) was set to 1.2 Å, molecular weight (MW) was set to 300–500, and rotatable bonds were set to 6–10. All of the hit reduction and hit screening parameters were applied against the PubChem database (https://pubchem.ncbi.nlm.nih.gov/) [33] identifying 53 candidate substances (Appendix A). *In silico* ADMET properties were determined for all 53 identified substance candidates, and, to narrow down the candidate substance selection, we used machine learning statistical methods and molecular docking.

### 4.5. Molecular Descriptors Calculation and Clustering

The 53 pharmacophores’ hit molecules further predicted chemical descriptors, structure comparisons, similarity searching, compound clustering, and data visualization by using the Marvin Beans tool [34] and ChemmineR: Cheminformatics Toolkit for R [35]. ChemmineR is a cheminformatics package for analyzing drug-like small molecule data in R (https://www.bioconductor.org/packages/release/bioc/vignettes/ChemmineR/inst/doc/ChemmineR.html). (Appendix A). It contains functions for efficient processing of large numbers of small molecules, physicochemical/structural property predictions, structural similarity searching, classification, and clustering of compound libraries with a wide spectrum of algorithms [36]. Compound structures are imported into ChemmineR in the generic structure definition file (SDF) format. The atom pair descriptors are calculated during the SDF import and stored in a searchable descriptor database as a list object [37]. Furthermore, we performed structural similarity searches against the generated atom pair descriptor databases by search function with the Tanimoto coefficient algorithm [38]. In addition to that, we performed structure-based clustering methods such as hierarchical clustering (Appendix A) and multidimensional scaling (MDS) clustering [38] (Appendix A). The algorithm uses single-linkage clustering to join compounds into similarity groups, where every member in a cluster shares with at least one another member a similarity value based on the specified threshold.

### 4.6. Statistical Machine Learning Methods 

In the present study, we applied discriminant algorithm (FDA), tree-based algorithm (C5.0), kernel-based algorithm (lsSVMrbf), ensemble algorithm (RF) [39], and k-Nearest Neighbors (kNN) [40] to develop classification prediction models for the training set (311 drug-like and 320 nondrug-like compounds) (Appendix A) [34], and the prediction performances of the models were measured with the testing set (53 hits) (Appendix A). Here, we give a brief overview of these statistical learning models. 

Flexible discriminant analysis (FDA) uses a nonparametric form of linear regression to handle LDA (Linear discriminant analysis) problem. Mixture discriminant analysis (MDA) models the density of each class from two or more Gaussian functions with different centroids [41,42]. C5.0 is an extension of the J48 algorithm and widely used decision tree algorithm. It is faster, more memory efficient, provides smaller decision trees, allows weight cases, sorts the useless features automatically, and supports boosting to improve the performance compared to J48 [42]. SVM (support vector machine) uses quadratic programming and Lagrange multipliers for this purpose. SVM applies kernel functions, including radial-basis function (RBF) and polynomial functions for nonlinear classification problems. Other kernel-based classifier least squares support vector machines (lsSVM) are special cases of SVM, which solve a linear system rather than using quadratic programming in optimizing the model parameters [42]. RF (Random forest) is chosen as the classifier [43,44] and ensemble learning method for classification, regression, and other tasks that operates by constructing a multitude of decision trees at training time and outputting the class that is the mode of the classes (classification) or mean prediction (regression) of the individual trees. An unbiased OOB (out-of-bag error) estimate [39], which is regarded as an excellent measure equivalent to cross-validation, can internally evaluate the generalization error of RF. k-NN [45,46] is a method for classifying test cases based on the majority voting principle in the feature space, or rather, if a sample has k-nearest neighbors, most of which belong to a certain category [47], it can be inferred that this sample also belongs to this category. The machine learning-based classification model was generated using the training set, and the quality of the generated model was assessed through a test set with tenfold cross-validation. The above-mentioned machine-learning algorithms classify compounds as drug-like and nondrug-like based on their molecular descriptors (Appendix A). 

### 4.7. Model Evaluation

The machine learning classification models applied in this study were evaluated by model evaluation parameters, such as accuracy rate (AR), sensitivity (SE), specificity (SP), positive predictive value (PPV), negative predictive value (NPV), detection rate (DR), balanced accuracy rate (bAR), F-score (FS), Matthews correlation coefficient (MCC), and Kappa statistic (κ). The specific calculation formulas are as follows:(1)$$\mathrm{Accuracy}\;\mathrm{Rate}\;(\mathrm{AR})\;=\frac{TP+TN}{TP+TN+FP+FN}$$
(2)$$\mathrm{Sensitivity}\;(\mathrm{SE})\;=\frac{TP}{TP+FN}$$
(3)$$\mathrm{Specificity}\;(\mathrm{SP})\;=\;\frac{TN}{TN+FP}$$
(4)$$\mathrm{Positive}\;\mathrm{Predictive}\;\mathrm{Value}\;(\mathrm{PPV})\;=\;\frac{TP}{TP+FP}$$
(5)$$\mathrm{Negative}\;\mathrm{Predictive}\;\mathrm{Value}\;(\mathrm{NPV})\;=\;\frac{TN}{TN+FN}$$
(6)$$\mathrm{Detection}\;\mathrm{Rate}\;(\mathrm{DR})\;=\;\frac{TP}{TP+TN+FN}$$
(7)$$\mathrm{balanced}\;\mathrm{Accuracy}\;\mathrm{Rate}\;(\mathrm{bAR})\;=\;\frac{SE+SP}{2}$$
(8)$$F-\mathrm{score}\;(\mathrm{FS})\;=\;2\frac{SE\;\times \;PPV}{SE+PPV}$$
(9)$$\mathrm{Matthews}\;\mathrm{correlation}\;\mathrm{coefficient}\;(\mathrm{MCC})\;=\;\frac{TP\;\times \;TN\;-\;FP\;\times \;FN}{\sqrt{\left(TP\;+\;FP\right)\left(TP\;+\;FN\right)\left(TN\;+\;FP\right)\left(TN\;+\;FN\right)}}$$
(10)$$\mathrm{Kappa}\;\mathrm{statistic}\;(\kappa )\;=\;\frac{AR\;+\;Pe}{1\;-\;Pe}$$
where *p_e_* = ((*TP* + *FN*) (*TP* + *FP*) + (*FP* + *TN*) (*FN* + *TN*))/*n*^2^, *TP* = true positives, *TN* = true negatives, *FP* = false positives, and *FP* = false negatives.

### 4.8. Molecular Docking

Molecular docking is widely used to estimate the binding affinity and analyze interactions between small molecules and proteins [28,48,49]. Here, we used AutoDock Vina v.4.2 [48,50,51] to bring the small molecules into the active site of the protein. The Lamarckian genetic algorithm was used as a number of individual population (150), max number of energy evaluation (25,000,000), max number of generation (27,000) [38], gene mutation rate (0.02), crossover rate (0.8), Cauchy beta (1.0), and GA window size (10.0). The grid box was set to the binding pocket of G6PD protein at X = 43.8621, Y = 15.1474, Z = 48.9404, and dimension (Å) at X = 54.8111, Y = 39.8676, Z = 43.0489 and exhaustiveness 8. The pose for given ligands identified on the basis of the highest binding energy. Only ligand’s flexibility was taken into account, and the protein was considered to be a rigid body. The resulting complexes were clustered according to their root mean square deviation (RMSD) values and binding energies (kcal/mol), which were calculated using the AutoDock scoring function.

### 4.9. Pharmacokinetics and Toxicity Risks Assessment

The SwissADME web tool (http://www.swissadme.ch/), available from the Swiss Institute of Bioinformatics (SIB), was used to calculate the physicochemical descriptors as well as to predict the ADME parameters, pharmacokinetic properties, drug-like nature, and medicinal chemistry friendliness of the selected compounds (Appendix A of top 5 compounds). The drug-likeness and ADMET (absorption, distribution, metabolism, elimination, and toxicity) properties were predicted by using an online Osiris property explorer (http://www.organic-chemistry.org/prog/peo/) (Appendix A). Drug-likeness was indicated by the Lipinski “Rule of 5′’ [52]. Lipinski’s rule of five was slightly modified according to the properties of existing small molecules. The restrictions were as follows: A logP ≤ 5.5, molecular weight ≤ 500D, and hydrogen bond acceptors (HBA) should be less than 10, hydrogen bond donors (HBD) should be less than 5, and rotated bonds must be less than 10. The molecules, which, in accordance with Lipinski’s rule, have good predicted activity, and good ADMET properties can be considered as the best drug molecule. Here, we performed the drug-likeness and ADMET properties for the top five best molecules. 

## 5. Conclusions and Future Directions

With the explosive growth of biomedical data, statistical machine learning methods have been increasingly used for the drug discovery. For identifying the drug-like or nondrug-like property of compounds, five machine learning models were used in this work. We built the pharmacophore model to screen novel and potent G6PD activators. The pharmacophore is composed of four chemical features, including two hydrogen bond donor spheres, one positive ionizable sphere, and one aromatic ring sphere. Thus, we have selected hits from the PubChem database mapping all chemical features of AG1 compound pharmacophore. The recognized hits were further assessed by statistical machine learning methods, docking, and in silico ADMET studies. The recognized hits may be utilized for creating novel and strong activators for G6PD. Moreover, the identified compounds can serve as a template for designing new G6PD agonists. These compounds can be considered as safe agents for further validation studies at the cell level, animal model, and even clinic setting. In conclusion, our study clearly shows that the virtual screening strategy combined with machine learning as well as structure-based molecular docking might be a powerful source for the identification of novel leads from chemical libraries. 

## Figures and Tables

**Figure 1 ijms-21-01523-f001:**
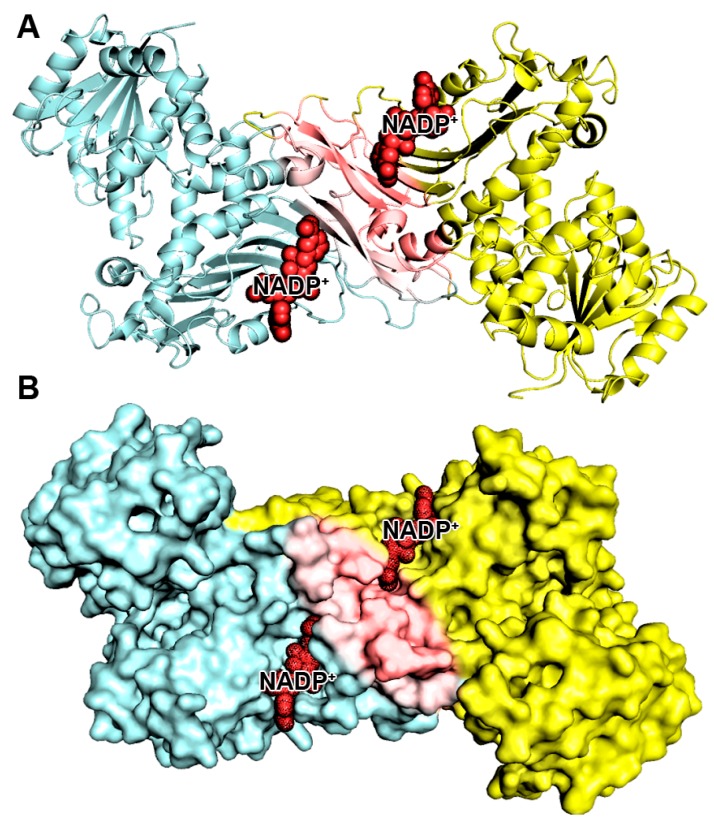
Depiction of the target enzyme G6PD and its active sites. (**A**) the target enzyme G6PD-dimer represented as a cartoon model; (**B**) the target enzyme G6PD-dimer represented as a surface model. Monomer-1 (cyan color) and monomer-2 (yellow color) are connected and formed dimer formation dimer interface active site (pink color), dark red color spheres designate the NADP^+^ binding sites. The dimer interface is designated with red.

**Figure 2 ijms-21-01523-f002:**
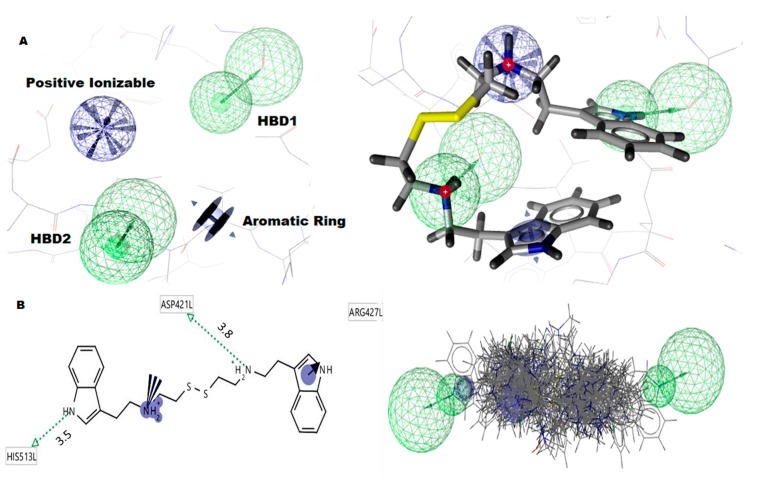
Pharmacophore model of G6PD-AG1-complex. (**A**) The pharmacophore model contains four pharmacophore features, such as two hydrogen donors (green color), one positive ionizable, and one aromatic ring (**B**). The G6PD-AG1 compound interacts with His513, ASP421, and ARG427 functional residues. The 53 hit molecules are fitted into pharmacophore features which are applied to the PubChem database.

**Figure 3 ijms-21-01523-f003:**
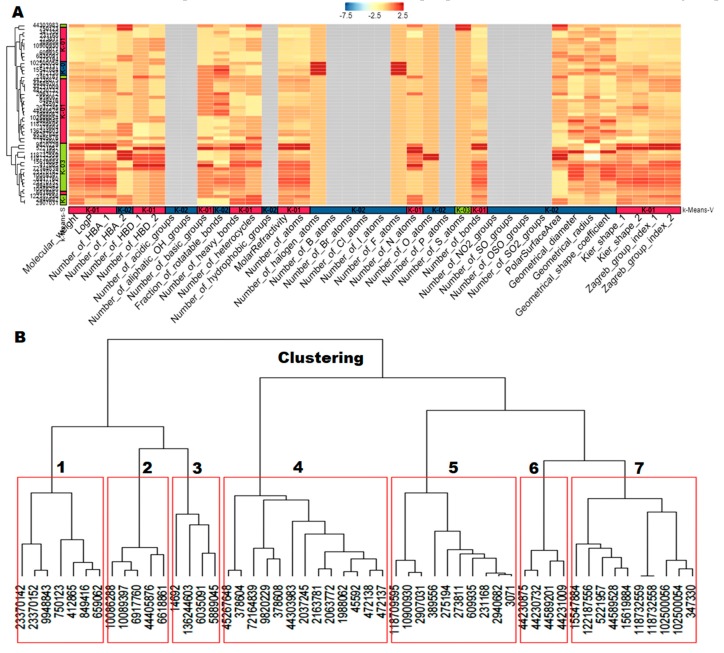
The molecular descriptors and clustering of 53 hit molecules. (**A**) The molecular descriptors were represented as a heatmap. It showed positive values as a red color and negative values as a blue color. (**B**) The molecular descriptors were classified as hierarchical clustering trees. The hierarchical clustering showed seven cluster trees. Cluster 1 has four compounds, Cluster 2 and Cluster 3 have ten compounds each, Cluster 4 has fourteen compounds, Cluster 5 has four compounds, Cluster 6 has five compounds, and Cluster 7 has seven compounds, respectively.

**Figure 4 ijms-21-01523-f004:**
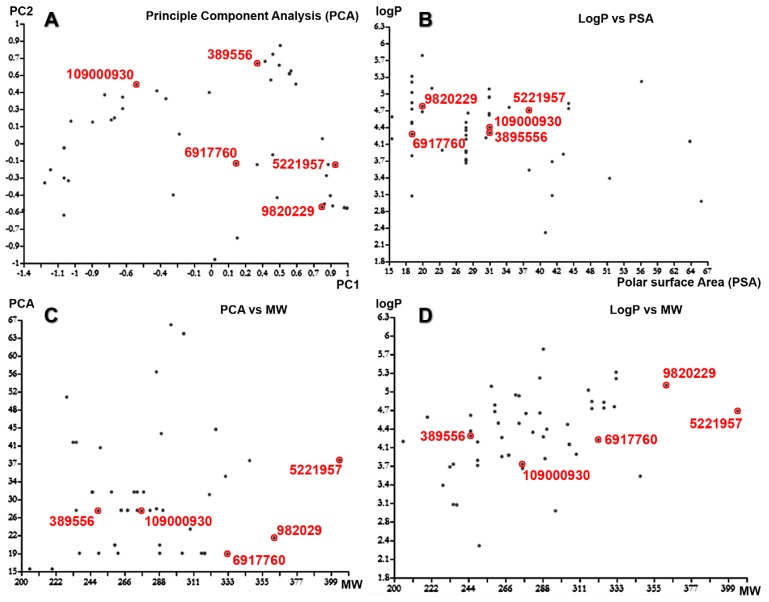
Principle component analysis (PCA) of 53 pharmacophores hit molecules. (**A**) The PCA showed various groups of compounds based on the Tanimoto coefficient (distance) between the first component (PCA1) against the second component (PCA2). (**B**) The logarithm of the calculated partition coefficient (logP) against the polar surface area (PSA) showed that the compounds have a maximum of 5.8 logP and 66 PSA. (**C**) The molecular weight (MW) against the PSA showed that the compounds have a maximum of 66 PSA and 400 MW. (**D**) The molecular weight (MW) against the logarithm of the calculated Partition coefficient (logP) showed that the compounds have a maximum of 5.8 logP and 400 MW.

**Figure 5 ijms-21-01523-f005:**
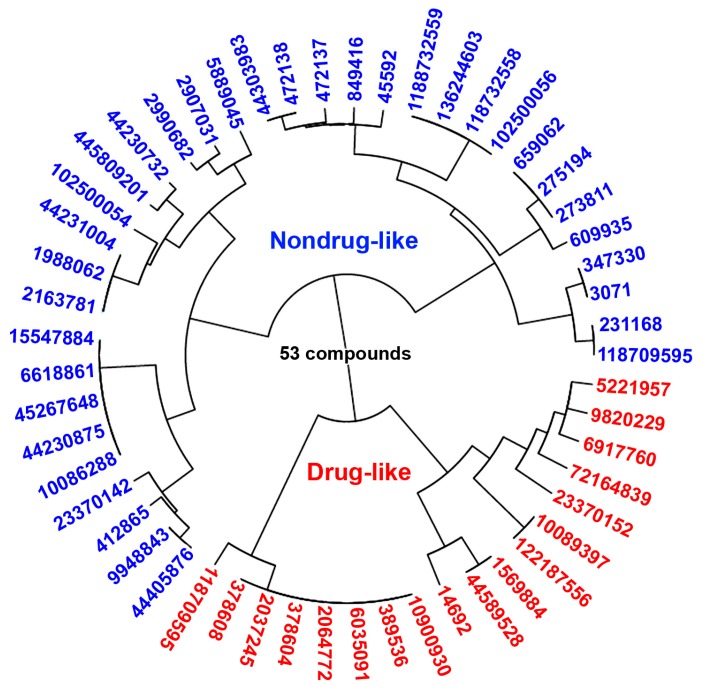
The statistical machine learning predictions and classified the 53 pharmacophore hit candidate molecules as a drug-like (18) and nondrug-like (35) compounds.

**Figure 6 ijms-21-01523-f006:**
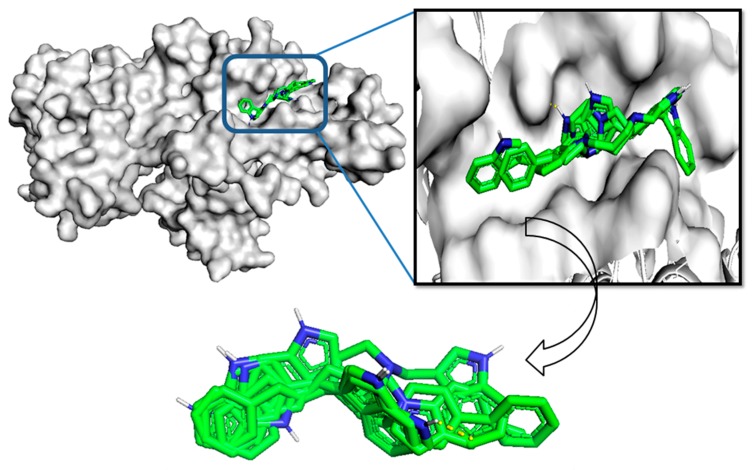
Molecular docking analysis illustrated all the drug-like (18) molecules docked into the G6PD-dimer interface active site (top). The binding energy of the top five compounds was aligned and superimposed (bottom).

**Figure 7 ijms-21-01523-f007:**
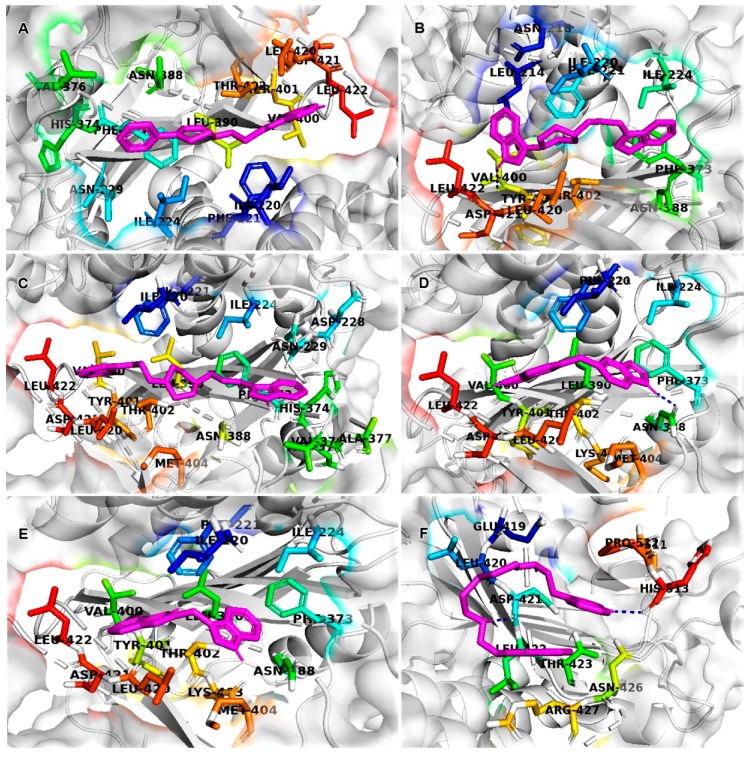
Protein–ligand interaction analysis of the best five compounds. (**A**) CID6917760 compound interacted with (ne–8.9 kcal/mole) active site of G6PD (dimer interface domain) functional residues; (**B**) CID9820229 compound interacted to (ne–7.6 kcal/mole) active site of G6PD (dimer interface domain) functional residues; (**C**) CID5221957 compound interacted with (ne–7.3 kcal/mole) active site of G6PD (dimer interface domain) functional residues; (**D**) CID389556 compound interacted to (ne–7.2 kcal/mole) active site of G6PD (dimer interface domain) functional residues; (**E**) CID10900930 compound interacted to (ne–7.0 kcal/mole) active site of G6PD (dimer interface domain) functional residues; (**F**) AG1 (CID6615809) compound interacted to (ne–6.1 kcal/mole) active site of G6PD (dimer interface domain) functional residues. The binding site functional residues represented as a sticks model with rainbow color; the best top five compounds were represented as a sticks model with magenta, and the G6PD protein represented as a cartoon model with white.

**Figure 8 ijms-21-01523-f008:**
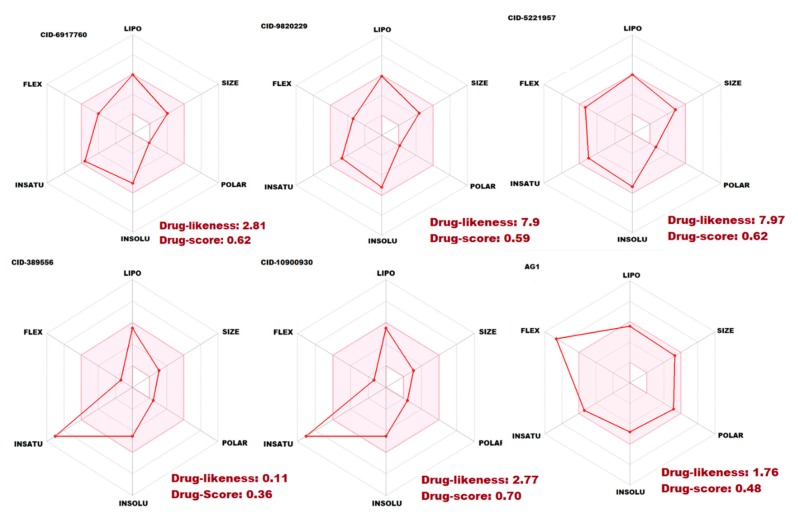
The ADMET properties of the five best G6PD small molecule activators CID6917760, CID9820229, CID5221957, CID389556, CID10900930, and AG1 (CID6615809). The pink area represents the optimal range for each properties (lipophilicity: XLOGP3 between ne−0.7 and +5.0, size: MW between 150 and 500 g/mol, polarity: TPSA between 20 and 130 Å^2^, solubility: log S not higher than 6, saturation: fraction of carbons in the sp3 hybridization not less than 0.25, and flexibility: no more than 9 rotatable bonds.

**Table 1 ijms-21-01523-t001:** Molecular docking scores (kcal/mol) and functional residues of the active molecules in the binding site of the protein G6PD.

PubChem IDs	Smiles Notation	Binding Energies (ΔG) Kcal/mol	Functional Amino Acids
**CID6917760**	C1CN(CC=C1C2=CC=CC=C2)CCCCC3=CNC4=CC=CC=C43	ne–8.9	ILE220, PHE221, ASN229, ASN388, ILE224, PHE373, TYR401, VAL400, THR402, ASP421, LEU420, LEU422
**CID9820229**	C1CN(CCC1C2=CNC3=CC=CC=C32)CCCN4CCC5=CC=CC=C54	ne–7.6	LEU214, PHE221, LEU420, THR402, ILE220, PHE373, ASN388, ILU224, LEU422
**CID5221957**	C1CN(CCN1CCCC2=CNC3=CC=CC=C32)CCCC4=CNC5=CC=CC=C54	ne–7.3	ASP421, LEU422, TYR401, LEU420, VAL400, THR402, PHE373, ILE224, ASN388, HIS374, ASP375, ASN229, VAL376, PHE221, ILE220
**CID389556**	C1=CC2=C(C=CN2)C=C1CC3=CC4=C(C=C3)NC=C4	ne–7.2	LEU420, VAL400, ASP421, TYR401, THR402, LEU422, PHE373, ASN388, ILE224, PHE221, ILE220
**CID10900930**	C1=CC=C2C(=C1)C=C(N2)CC3=CNC4=CC=CC=C43	ne–7.0	PHE373, THR402, LEU422, TYR401, ASP521, LEU420, VAL400, PHE221, ILE220, ASN388
**CID6615809** **(AG1)**	C1=CC=C2C(=C1)C(=CN2)CCNCCSSCCNCCC3=CNC4=CC=CC=C43	ne–6.1	LEU420, THR423, ASN426, ASP421, ARG427, LEU422

**Table 2 ijms-21-01523-t002:** Physicochemical properties, lipophilicity, water-solubility, pharmacokinetics, drug-likeness, and medicinal chemistry properties of selected ligands determined by the SwissADME server.

Descriptors	CID6917760	CID9820229	CID5221957	CID389556	CID10900930	AG1
**Physicochemical Properties**						
Formula	C23H26N2	C24H29N3	C26H32N4	C17H14N2	C17H14N2	C24H30N4S2
Molecular weight	330.47 g/mol	359.51 g/mol	400.56 g/mol	246.31 g/mol	246.31 g/mol	438.65 g/mol
Num. heavy atoms	25	27	30	19	19	30
Num. arom. heavy atoms	15	15	18	18	18	18
Fraction Csp3	0.03	0.42	0.38	0.06	0.06	0.33
Num. rotatable bonds	6	5	8	2	2	13
Num. H-bond acceptors	1	1	2	0	0	2
Num. H-bond donors	1	1	2	2	2	4
Molar Refractivity	111.22	121.13	134.38	79.61	79.61	134.04
TPSA	19.03 Å^2^	22.27 Å^2^	38.06 Å^2^	31.58 Å^2^	31.58 Å^2^	106.24 Å^2^
**Lipophilicity**						
Log P_o/w_ (iLOGP)	3.60	3.66	3.66	2.07	2.21	3.99
Log P_o/w_ (XLOGP3)	4.93	4.84	4.95	4.10	4.13	4.22
Log P_o/w_ (WLOGP)	4.90	4.04	4.07	4.24	4.24	5.00
Log P_o/w_ (MLOGP)	4.16	3.83	3.24	3.00	3.00	2.83
Log P_o/w_ (SILICOS-IT)	5.70	4.92	5.90	5.01	5.01	6.29
Consensus Log P_o/w_	4.66	4.26	4.36	3.68	3.72	4.47
**Water Solubility**						
Log S (ESOL)	ne–5.04	ne–5.20	ne–5.36	ne–4.52	ne–4.54	ne–4.80
Solubility	2.99 × 10^−3^ mg/mL; 9.06 × 10^−6^ mol/L	2.27 × 10^−3^ mg/mL; 6.32 × 10^−6^ mol/L	1.76 × 10^−3^ mg/mL; 4.39 × 10^−6^ mol/L	7.45 × 10^−3^ mg/mL; 3.03 × 10^−5^ mol/L	7.14 × 10^−3^ mg/mL; 2.90 × 10^−5^ mol/L	6.88 × 10^−3^ mg/mL; 1.57 × 10^−5^ mol/L
Class	Moderately soluble	Moderately soluble	Moderately soluble	Moderately soluble	Moderately soluble	Moderately soluble
Log S (Ali)	ne–5.07	ne–5.04	ne–5.49	ne–4.47	ne–4.50	ne–6.16
Solubility	2.83 × 10^−3^ mg/mL; 8.58 × 10^−6^ mol/L	3.27 × 10^−3^ mg/mL; 9.09 × 10^−6^ mol/L	1.30 × 10^−3^ mg/mL; 3.26 × 10^−6^ mol/L	8.37 × 10^−3^ mg/mL; 3.40 × 10^−5^ mol/L	7.79 × 10^−3^ mg/mL; 3.16 × 10^−5^ mol/L	3.03 × 10^−4^ mg/mL; 6.90 × 10^−7^ mol/L
Class	Moderately soluble	Moderately soluble	Moderately soluble	Moderately soluble	Moderately soluble	Poorly soluble
Log S (SILICOS-IT)	ne–7.87	ne–7.35	ne–8.82	ne–7.10	ne–7.10	ne–10.13
Solubility	4.49 × 10^−6^ mg/mL; 1.36 × 10^−8^ mol/L	1.61 × 10^−5^ mg/mL; 4.48 × 10^−8^ mol/L	6.09 × 10^−7^ mg/mL; 1.52 × 10^−9^ mol/L	1.97 × 10^−5^ mg/mL; 8.00 × 10^−8^ mol/L	1.97 × 10^−5^ mg/mL; 8.00 × 10^−8^ mol/L	3.21 × 10^−8^ mg/mL; 7.33 × 10^−11^ mol/L
Class	Poorly soluble	Poorly soluble	Poorly soluble	Poorly soluble	Poorly soluble	Insoluble
**Pharmacokinetics**						
GI absorption	High	High	High	High	High	High
BBB permeant	Yes	Yes	Yes	Yes	Yes	No
P-gp substrate	Yes	Yes	Yes	Yes	Yes	Yes
CYP1A2 inhibitor	Yes	Yes	Yes	Yes	Yes	Yes
CYP2C19 inhibitor	Yes	No	No	Yes	Yes	Yes
CYP2C9 inhibitor	No	No	No	No	No	No
CYP2D6 inhibitor	Yes	Yes	Yes	Yes	Yes	Yes
CYP3A4 inhibitor	Yes	Yes	Yes	Yes	Yes	Yes
Log K_p_ (skin permeation)	ne–4.82 cm/s	ne–5.06 cm/s	ne–5.23 cm/s	ne–4.89 cm/s	ne–4.87 cm/s	ne–5.98 cm/s
**Drug likeness**						
Lipinski	Yes; 1 violation: MLOGP > 4.15	Yes; 0 violation	Yes; 0 violation	Yes; 0 violation	Yes; 0 violation	Yes; 0 violation
Ghose	Yes	Yes	No; 1 violation: MR > 130	Yes	Yes	No; 1 violation: MR > 130
Veber	Yes	Yes	Yes	Yes	Yes	No; 1 violation: Rotors > 10
Egan	Yes	Yes	Yes	Yes	Yes	Yes
Muegge	Yes	Yes	Yes	Yes	Yes	Yes
Bioavailability Score	0.05	0.05	0.55	0.55	0.55	0.55
**Medicinal Chemistry**						
PAINS	0 alert	0 alert	0 alert	0 alert	0 alert	0 alert
Brenk	0 alert	0 alert	0 alert	0 alert	0 alert	1 alert: disulphide
Lead likeness	No; 1 violation: XLOGP3 > 3.5	No; 2 violations: MW > 350, XLOGP3 > 3.5	No; 3 violations: MW > 350, Rotors > 7, XLOGP3 > 3.5	No; 2 violations: MW < 250, XLOGP3 > 3.5	No; 2 violations: MW < 250, XLOGP3 > 3.5	No; 3 violations: MW > 350, Rotors > 7, XLOGP3 > 3.5
Synthetic accessibility	3.24	3.31	2.81	1.70	2.20	3.17

**Table 3 ijms-21-01523-t003:** The best five compounds of ADMET properties are calculated by Osiris molecular property explorer.

Properties	CID6917760	CID9820229	CID5221957	CID389556	CID10900930	AG1
Mutagenic	No	No	No	Partial	No	No
Tumorigenic	No	No	No	Partial	No	No
Irritant	No	No	No	No	No	No
Reproductive effect	No	No	No	No	No	No
cLogP	4.77	4.76	4.63	3.55	3.61	4.22
Solubility	ne–3.99	ne–4.41	ne–3.84	ne–4.38	ne–4.40	ne–5.18
MW	330	359	400	246	246	438
TPSA	19.03 Å^2^	22.27 Å^2^	38.06 Å^2^	31.58 Å^2^	31.58 Å^2^	106.2 Å^2^
Drug likeness	2.81	7.9	7.97	0.11	2.77	1.76
Drug score	0.62	0.59	0.62	0.36	0.70	0.48

## Data Availability

The crystal structure of G6PD in complex with structural NADP (PDB ID: 2BHL) was downloaded from the Protein Data Bank (PDB) (https://www.rcsb.org/structure/2BHL). The G6PD activator AG1 analogs were collected from PubChem database libraries. The current study output files and machine learning sets are included in the supplementary data. The most recent version of the source code used in this study is available from GitHub: https://github.com/madhubioinformatics/Discovery-of-small-molecule-activators-for-G6PD-using-machine-learning-approaches.

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
