# Peer review of "Discovery of Small-Molecule Activators for Glucose-6-Phosphate Dehydrogenase (G6PD) Using Machine Learning Approaches"

_ijms, 2020, doi:10.3390/ijms21041523_

Round 1
Reviewer 1 Report
x
Author Response
We appreciate the positive opinion regarding our work by reviewer 1.
Reviewer 2 Report
This manuscript aims to discover therapeutic molecules to treat G6PD deficiency, using machine learning methods. I believe that combining machine learning methods with traditional structure-based virtual screening, as authors performed in that manuscript, is a next-generation strategy that possess grave importance for drug discovery. I have no major concerns about the paper. The only minor addition might be extending the discussion on the properties of five hits and explicitly stating their biological effects. Their chemical properties are well defined, but in order to make the paper more accessible for clinicians and biologist, authors should include a brief paragraph about the biology of this molecules.
Author Response
We highly appreciate the reviewer for valuable suggestions and inputs for improving the manuscript. We have gone through the text again to improve the language. We have updated the properties of hit compounds and their biological effects and biology of these compounds in the discussion section. Please refer to the traced version of the manuscript for specific changes.
Reviewer 3 Report
1) General comments
Very extensive editing of the English language and style is needed. There are multiple grammar errors (many extremely trivial like "features are play a vital"). The structure of the text should be improved. Some parts need to be re-written (e.g., the first part of abstract that is "an introduction" is too long and needs to be shortened). In general the whole abstract should be re-written (multiple grammar errors, too little focus on results, etc.).
All data and intermediate files need to be provided (this means: PDB models before and after each step; predictions of bioinformatics tools such as CASTp, LigandScout, etc.; datasets for ML, their vectors, models, and predictions from SVM, RF, etc.). The files should be added as supplementary material or stored in the external databases (e.g., FigShare) as separate files. Do not use program-specific formats. Without the files, it is impossible to validate the correctness of the bioinformatic results. Until access to the raw data is not possible, all results presented in the manuscript are considered as "not confirmed by the data". The current version of the section "Data availability", I consider as a joke.
2) Specific comments for revision:
a) major:
- the whole ML part is very disputable (the authors do not show ML results except Fig 5. that is encoded - PubChem id not provided, fix that). The datasets used for training need to provided (in ref 20 they are missing). The ML models and predictions for individual train and test targets need to be provided and summarized in the supplementary table. You mention in "Methods" a huge list of metrics e.g. MCC, F1 and you never use them. In the end, the whole ML is pointless as you choose 5 best candidates using binding energies.
- what is the relationship of Fig. 3b, Fig 5 and Table 1. (for Fig 5 use horizontal tree presentation such as in 3b, use PubChem ids and mark there top 5 + AG1)?
b) minor
- the reference list is corrupted, it has 31 positions (with doubled numbering) while in the text you cite at least 48 papers.
- do not use acronyms without explaining them (e.g. RBCs)
- always use units e.g. Å2 for surface areas
- Fig. 3 and Fig 5- can you see any pattern from the clustering or ML predictions (this should be discussed extensively)
- Fig. 1 coloring could be improved e.g. shades of cyan and yellow for the interface, marking domains, etc.)
- Fig 2b the distance to interacting residues (green lines) should be provided
- Fig 4 add numbers from 1-53 instead dots (add a supplementary table with mapping numbers to PubChem ids)
- Fig 6 (bottom model) is not essential and should be replaced by the structures of the top 5 compounds, AG1, and their alignment), the compound coloring in the insert should be improved
- Fig 7 - do something with the coloring, the pictures should be less Claude_Monet+van_Gogh. The legend should be "Protein-ligand interaction analysis of best five compounds and AG1." - you have six panels here (the same for Table 1)
- Fig 8 - the structures (small icons) should be in the same scale. AG1 panel is duplicated. Add some score (e.g. the area within favorable space)
- Table 3 should be extended in the supplement to all 53 targets (can be CSV file for simplicity).
Author Response
General comments:
- Very extensive editing of the English language and style is needed. There are multiple grammar errors (many extremely trivial, like "features are play a vital"). The structure of the text should be improved. Some parts need to be re-written (e.g., the first part of abstract that is "an introduction" is too long and needs to be shortened). In general, the whole abstract should be re-written (multiple grammar errors, too little focus on results, etc.).
Response: We appreciate the suggestions for language improvement. We have revised and updated the manuscript. We rewrote the text to provide a smoother narrative. The final version of the revised manuscript was proofread by two native English speakers.
- All data and intermediate files need to be provided (this means: PDB models before and after each step; predictions of bioinformatics tools such as CASTp, LigandScout, etc.; datasets for ML, their vectors, models, and predictions from SVM, RF, etc.). The files should be added as supplementary material or stored in the external databases (e.g., FigShare) as separate files. Do not use program-specific formats. Without the files, it is impossible to validate the correctness of the bioinformatic results. Until access to the raw data is not possible, all results presented in the manuscript are considered as "not confirmed by the data."
Response: In the revised version of the manuscript, we have added Supplemental data with Supplementary Figure 1: CASTp active site prediction of the G6PD molecule and its features. Supplementary Figure 2: Molecular structure of 53 compounds candidates selected for machine learning analysis. A raw chemical table file is attached as a supplementary file 04_53_hits.sdf
Table S1. Training data set used in the study. This data set contains 631 compounds and six molecular descriptors.
Table S2. Test data set used in the study. This data set contains 53 compounds and six molecular descriptors.
As well as the set of individual supplementary files: 01_Before_processing_G6PD.pdb; 02_After_processing_G6PD.pdb; 03_Pharmacophore_complex.pdb; 04_53_hits.sdf; 05_R-source_code.txt; 06_ML_results.xlsx; 07_Heatmap_descriptors.xlsx; 08_SWISS-ADMET_of_top_5_compounds.xlsx; 09_AMDET_properites_of_53_compounds.xlsx
That comprehensively covers all the aspects of the compounds' discovery and evaluation process at each step. We added references to the supplementary figures, tables, and individual files at every step of the main manuscript text.
The current version of the section "Data availability," I consider as a joke.
Response: We acknowledge the deficiency of our data availability section. In the revised version of the manuscript, we provided all study intermediate and source files, as well as the machine learning sets and the R-source code. The current data availability is stated as follows:
The crystal structure of G6PD in complex with structural NADP (PDB ID: 6E08) was downloaded from the Protein Data Bank (PDB). The G6PD activator AG1 analogs were collected from PubChem database libraries. The current study output files and machine learning sets are included in the supplementary data. The most recent version of the source code used in this study is available from GitHub: https://github.com/madhubioinformatics/Discovery-of-small-molecule-activators-for-G6PD-using-machine-learning-approaches. The PDB files that support the findings of the study are available from Protein Data Bank under ascension number 2BHL: https://www.rcsb.org/structure/2BHL
Specific comments for revision
Major
- The whole ML part is very disputable (the authors do not show ML results except Fig 5. that is encoded - PubChem id not provided, fix that). The datasets used for training need to provided (in ref 20 they are missing). The ML models and predictions for individual train and test targets need to be provided and summarized in the supplementary table. You mention in "Methods" a huge list of metrics, e.g., MCC, F1, and you never use them.
Response: As per the reviewer's request here, we provide all machine learning results as well as training and testing datasets with a summary as the supplementary tables and additional files. We fixed the PubChem id instead of the numbering of compounds. In this current study, we applied the machine learning process reported by Korkmaz et al., 2015 to our pharmacophore hit molecules and classified into drug and non-drug molecules. We have also updated the missing references and missing information in the revised manuscript, hoping that the reviewer will find a comprehensive description of the manuscript methodology.
Korkmaz S, Zararsiz G, Goksuluk D (2015) MLViS: A Web Tool for Machine Learning-Based Virtual Screening in Early-Phase of Drug Discovery and Development. PLoS ONE 10(4): e0124600.
- In the end, the whole ML is pointless as you choose 5 best candidates using binding energies.
Response: In this study, we applied machine learning methods reported by Korkmaz et al., 2015 to our pharmacophore hit molecules and classified using ML into drug-like and non-drug like molecules. Then we took drug-like molecules produced by ML and applied established bioinformatics pipeline and selected top five candidates among drug-like molecules based on binding energies and in silico ADMET properties. Therefore, the machine learning algorithm is a crucial step to separate drug-like and non-drug like substances, the final grading by binding energies. By omitting ML step substances with high binding energies, but not suitable for pharmacological use such as due to toxicity, mutagenic properties, etc., would be identified, dramatically increasing the number of candidates for in vitro and in vivo validation. Thus, we respectfully disagree with the reviewer's opinion that the ML step was pointless.
- What is the relationship between Fig. 3b, Fig 5, and Table1. (for Fig 5 use horizontal tree presentation such as in 3b, use PubChem ids and mark there top 5 + AG1)?
Response: We have calculated the molecular descriptors and visualized (heatmap) of 53 pharmacophores hit molecules then performed clustering analysis. The multidimensional scaling (MDS) clustering use single-linkage clustering (7 cluster tree) to join compounds into similarity groups, where every member in a cluster shares with at least one another member a similarity value based on a specified threshold (Fig3a and 3b). Then, we have taken six molecular descriptors and applied machine learning methods reported by Korkmaz et al., 2015 reported all to our pharmacophore hit molecules and classified into drug-like and non-drug like molecules (Fig.5). Finally, we took drug-like compounds and applied established bioinformatics pipeline and selected the top five candidates among drug-like molecules based on binding energies (Table 1) and in silico ADMET properties. We have also added PubChem ID to figure 5. We produced the horizontal tree presentation for figure 5, but the overall design was not working well with the page space limitation, as the figure required more space and the elements look small, spherical Figure 5 using the space of the page more efficiently and in our opinion delivering well it’s intended message to the reader of machine learning separation of the compounds. If the reviewer still disagrees and believes the tree-like structure would be more beneficial to the reader, we will introduce the tree-like structure to figure 5.
Minor Concerns
- the reference list is corrupted; it has 31 positions (with doubled numbering) while in the text you cite at least 48 papers.
Response: We apologize for the Endnote software version mismatch issue; we have corrected all references and removed the duplicated references in the revised manuscript. The current list of references consists of 52 and presented in the main manuscript file without an issue.
- Do not use acronyms without explaining them (e.g., RBCs), always use units, e.g., Å2 for surface areas.
Response: We have revised and updated the acronyms words and units in the manuscript.
- Fig. 3 and Fig 5- can you see any pattern from the clustering or ML predictions (this should be discussed extensively).
Response: The Fig.3 clustering designed by all molecular descriptors of 53 pharmacophores hit molecules (please refer to the major comment 3 response). The machine learning clustering designed by six molecular descriptors of 53 pharmacophores hit molecules (experimental set) compared by ML algorithm to the training set data presented by Korkmaz et al., 2015. We have edited the discussion and materials and methods in the revised manuscript to avoid confusion.
- Fig. 1 coloring could be improved, e.g., shades of cyan and yellow for the interface, marking domains, etc.).
Response: Thanks for reviewer suggestion, we have revised the figure for clarity and improved colors. We designated the dimer interface with the red color. We are happy to revise the figure further if the reviewer finds figure 1 still lacking clarity.
- Fig 2b, the distance to interacting residues (green lines), should be provided.
Response: We appreciate the reviewer for his suggestion; we have corrected the Fig.2b distance of interacting residues in the revised version of the manuscript.
- Fig 4 adds numbers from 1-53 instead of dots (add a supplementary table with mapping numbers to PubChem ids).
Response: We appreciate the reviewer for his suggestion, given the space limitation of the graph, it would be difficult to put all 53 PubChem ids, as the ids will overlap and make the figure difficult to understand. However, we provided the top five best compounds PubChem ids in Fig.4.
- Fig 6 (bottom model) is not essential and should be replaced by the structures of the top 5 compounds, AG1, and their alignment), the compound coloring in the insert should be improved.
Response: We have updated the Fig.6 as per reviewer suggestion.
- Fig 7 - do something with the coloring, the pictures should be less Claude_Monet+van_Gogh. The legend should be "Protein-ligand interaction analysis of best five compounds and AG1." - you have six panels here (the same for Table 1)
Response: We have revised the figure and adjusted the color scheme of the Fig.7 for clarity. Specifically presenting the background as grayscale.
- Fig 8 - the structures (small icons) should be on the same scale. AG1 panel is duplicated. Add some score (e.g., the area within favorable space)
Response: We have revised the Fig.8 as per reviewer suggestion.
- Table 3 should be extended in the supplement to all 53 targets (can be CSV file for simplicity).
Response: We have provided all 53 compounds of in silico ADMET properties in the supplementary data as a separate file 09_AMDET_properites_of_53_compounds.xlsx. For table 3, we have applied machine learning methods and classified the testing set as drug-like (18 compounds) and non-drug like (35 compounds). We chose drug-like (18 compounds) compounds for bioinformatics pipeline analysis and selected five best binding energy compounds and performed in silico ADMET properties (Table.3).
Round 2
Reviewer 3 Report
1) The manuscript in the current form is not acceptable as it contains changes in "Track Change Mode" that had not been validated/accepted by the authors. It was the authors' responsibility to check and accept (or not) the corrections after proofreading by a native speaker (some corrections made by the native speaker(s) are not adequate due to their lack of understanding of the subject). The same with the figures (old versions should be removed from the final file).
2) the supplementary files in xlsx should be converted to csv format
3) Supplementary Tables 1 & 2 should be csv files (there is no added value to present them as *.docx files, actually it makes harder to process them).
Author Response
1) The manuscript in the current form is not acceptable as it contains changes in "Track Change Mode" that had not been validated/accepted by the authors. It was the authors' responsibility to check and accept (or not) the corrections after proofreading by a native speaker (some corrections made by the native speaker(s) are not adequate due to their lack of understanding of the subject). The same with the figures (old versions should be removed from the final file).
Response: We agree with the reviewer, and surprised that the final manuscript pdf was from the traced version rather than from a clean version that is submitted as the main manuscript doc file. It might be a file name mistake during the upload on our side, and we apologize for this technical issue and any confusion that it might have caused. The traced version represents the difference of the revised version from the initial submission. We did verify all the corrections from the native speakers proofreading.
2) the supplementary files in xlsx should be converted to CSV format
Response: We appreciate the good suggestions, for muli-platfrom compatibility of supplementary data. We have converted all the tables into CSV format as suggested.
3) Supplementary Tables 1 & 2 should be csv files (there is no added value to present them as *.docx files, actually it makes harder to process them).
We agree with the reviewer and moved Tables 1, 2 and 3 to supplementary csv files. Indeed Office package is struggling to process docx files with large tables and as the reviewer pointed out there is no added value in doing this.